# Is the Urban Form a Driver of Heavy Metal Pollution in Road Dust? Evidence from Mexico City

Anahi Aguilera [1,2], Dorian Bautista-Hernández [2,*], Francisco Bautista [2], Avto Goguitchaichvili [3] and Rubén Cejudo [3]

1    Posgrado en Ciencias Biológicas, Universidad Nacional Autónoma de México, Morelia 58341, Michoacán, Mexico; aaguilera@cieco.unam.mx

2    Laboratorio Universitario de Geofísica Ambiental, Centro de Investigaciones en Geografía Ambiental, Universidad Nacional Autónoma de México, Morelia 58341, Michoacán, Mexico; leptosol@ciga.unam.mx

3    Laboratorio Universitario de Geofísica Ambiental, Instituto de Geofísica Unidad Michoacán, Universidad Nacional Autónoma de México, Morelia 58341, Michoacán, Mexico; avto@geofisica.unam.mx (A.G.); ruben@igeofisica.unam.mx (R.C.)

\*    Correspondence: dobautistah@gmail.com

**Abstract:** Environmental pollution is a negative externality of urbanization and is of great concern due to the fact that it poses serious problems to human health. Pollutants, such as heavy metals, have been found in urban road dust; however, it is unclear whether the urban form has a role in its accumulation, mainly in cases where there is no dominant unique source. We collected 482 samples of road dust, we determined the concentrations of five heavy metals (Cr, Cu, Pb, Zn, and Ni) using inductively coupled plasma optical emission spectrometry (ICP-OES), and then we derived the pollution load index (PLI). After estimating the mostly anthropogenic origin of these pollutants based on global levels of reference, there were two main aims of this study. Firstly, to analyze the spatial correlation of heavy metals, and secondly, to identify the main factors that influenced the heavy metal concentrations in the road dust of Mexico City. We did this by using a spatial autocorrelation indicator (Global Moran's I) and applying ordinary least squares (OLS) and spatial regression models. The results indicated low levels of positive spatial autocorrelation for all heavy metals. Most variables failed to detect any relationship with heavy metals. The median strip area in the roads had a weak (significance level of 90%) but consistent positive relationship with Cr, Cu, Ni, Pb, and the PLI. The distance to the airport had a weak (significance level of 90%) and inverse relationship with Pb. Manufacturing units were associated with an increase in Cu (significance level of 95%), while the entropy index was associated with an increase in Ni (significance level of 95%).

**Keywords:** road dust; heavy metals; Mexico City; urban pollution; urban form

## 1. Introduction

The world is becoming increasingly urban. In the developing world, Latin America has already achieved this transition toward an urbanized society. For example, in Mexico, it is estimated that 80% of the population lives in urban areas [1]. The capital, Mexico City, along with its metropolitan area, concentrates around 17.5% of the country's population and has over 40,000 industries and 4 million vehicles that consume more than 40 million liters of fossil fuels per day, releasing, as a result, thousands of tons of pollutants into the urban environment [2]. Environmental pollution is one of the main negative externalities of huge urban agglomerations, especially in the developing context where weak institutions and planning efforts aggravate the problem [3].

Air pollution has attracted a great deal of attention, as it is considered one of the main causes of death in cities [4]; therefore, air pollution has been widely researched. Less attention has been paid to road dust pollution [5]; however, we assume that they are likely interlinked. Research reported that road dust is a sink for polluting emissions,

which are deposited on the surface of streets, sidewalks, and windows [6]. At the same time, road dust is a source of pollutants of atmospheric particulate matter [7]. Studies found that urban structure factors, such as land use, industrial development, and building construction, worsened the pollution in urban areas [8,9]. From a social point of view, aspects such as tax revenue and education level are associated with a decrease in urban pollution [9]. In terms of country-level data, researchers have tested the environmental Kuznets curve (EKC) hypothesis, which states that the relationship between gross domestic product (GDP) per capita and different environmental indicators exhibit an inverted-U curve [10,11].

Heavy metals are some of the major pollutants found in road dust [12–14]. Due to their abundance, toxicity, persistence, and bioaccumulation, heavy metals can cause permanent damage to ecosystems and humans [15]. The severity of health problems due to heavy metal toxicity depends on several factors, such as the type and form of the element, the route and duration of exposure, and to a greater extent, the susceptibility of each person [16]. Low concentrations of non-essential heavy metals (As, Hg, Pb, Cr, and Cd) can be lethal to animals. In Mexico City, the median levels of lead in children's blood were found to be close to the reference level for public health interventions (5.0 µg/dL) [17]. Prenatal lead exposure has been associated with a decrease in child growth [18]. Even essential metals (Zn, Cu, and Ni), required for the proper function of different enzymes, can become toxic in high concentrations, inducing the generation of reactive nitrogen and oxygen species. This can result in the peroxidation of lipids, as well as the functional deterioration of DNA and proteins [19]. Nickel diminishes the protection that taurine provides against neurodegeneration [16].

The sources of heavy metals in road dust can be diverse. In some cities, the origins are very specific to major stationary emitting sources; for example, the smelting industry [13], e-waste recycling [20,21], and mining activities [22]. Natural processes may also be the cause of an increase in the concentrations of trace metals [23]. Using pollutant-tracing approaches, mobile sources of heavy metals have also been identified, such as vehicular emissions [24]. It is considered that Cu, Pb, Zn, and Cr are traffic-related metals [25–27]. The quantities emitted vary, but some metals can be linked to specific vehicle parts. For example, Cu emissions are generally related to brake abrasion [28,29]; Zn is mainly emitted by tire wear [25], as well as from diesel exhaust emissions [27]. Some Zn compounds are used as additives for motor oil [30]. In the case of Cr, this metal can originate from exhaust emissions [26]. Pb can come from brakes and the loss of lead wheel weights [27]; Cr and Pb have been reported to originate also from yellow street paint which contains lead chromate. Paint and pigments crack, peel, and turn to chalk, mobilizing metal particles into the urban environment [31,32]. Ni can come from mixed industrial/fuel-oil combustion [30].

In cities with an economy mostly related to the service sector, there are multiple small and scattered possible sources, which can be mobile or fixed. Therefore, it is vital to determine what characteristics of the urban form act as driving factors of heavy metal pollution in road dust to obtain a better understanding of the cycle in the city. This would bring insights to help plan measures that limit the exposure of the population to these pollutants.

This work goes beyond the description of heavy metals in urban road dust. To the best of our knowledge, this is the first attempt to make systematic statistical inferences regarding the characteristics of the urban form that could influence the concentration of heavy metals in urban road dust. In this paper, we respond to two main questions: (1) Is there a spatial correlation in the heavy metal contents of points sampled across Mexico City? (2) What are the main factors that explain the distribution and concentration of heavy metals in urban road dust? The first question is addressed by applying the Moran's I spatial correlation test to the heavy metal concentrations, and the second question by using linear regression models to analyze the relationship between heavy metals and the urban form.

## 2. Literature Review

There have been several approaches to evaluate the effects of the built environment on urban pollution, mainly in the air and taking the city as the unit of analysis. Liang et al. [9] in the Beijing–Tianjin–Hebei urban agglomeration, using a geographically weighted regression model, found that the rate of urbanization, the formation of metropolises, and the level of economic and industrial development, as well as building and road construction aggravate the environmental pollution (measured as an index for air, soil, and water pollution).

Predictors associated with a decrease in urban pollution include the industrial level, tax revenue, education level, and the use of the internet. Implementing taxes to protect the environment promotes the modernization of highly polluting industries, while an increase in resident incomes promotes a shift in the regional economy to low-pollution, knowledge-based industries. An improvement in the educational level drives environmental protection technology, improving the environmental quality.

Jung et al. [8], in Korea, applying a Bayesian spatial regression model, found that the total population, the commercial area, the industrial area, the total area, and the gross domestic product per person are factors associated with worse air pollution (nitrogen oxides, sulfur oxides, PM10, and PM2.5). Zhang et al. [33] in Calgary, Canada, using a land-use regression model, found that the main factors that increased the heavy metal concentrations in airborne particulate matter were industrial point sources.

Industrial and commercial zoning, as well as traffic indicators and population density, were also good predictors for most elements. In the case of specifically addressing the pollution by heavy metals in road dust, Alharbi et al. [14] compared the spatial distribution in two metropolitan cities (a typical corridor city and a compact city) of Saudi Arabia. They found that centrality was an important factor for determining the spatial distribution of heavy metals in road dust, which increased in concentration gradually toward the city center.

In Mexico City, research demonstrated that greenhouse gas emissions, caused by commuting, can be reduced by increasing the mix of residential and economic uses, as well as concentrating jobs near employment centers and economic activity corridors [34]. Studies determined that the sources of heavy metals in the urban road dust in Mexico City must be anthropogenic [35]. Therefore, we hypothesized that factors related to industrial land use, mixed land-use, and job density could be related to the concentrations of heavy metals associated with traffic, like Pb, Cu, and Zn. We hypothesized a negative association with the urban vegetation cover assuming a depuration effect.

## 3. Study Site

The Mexico City metropolitan area had a population of around 21,000,000 inhabitants in 2015 and was among the four largest urban agglomerations in the world [36]. The metropolitan area is formed by the urban areas of three states: Mexico City, the State of Mexico, and Hidalgo. Mexico City comprises mostly the central and south parts of the metropolitan area. The metropolitan area is located in a valley at 2240 m.a.s.l. The unique topography, with mountains surrounding the metropolis, results in thermal inversions preventing the dispersion of pollutants during the winter season from November to April.

Sierra del Ajusco's mountains in the southwest prevent the passage of the prevailing winds (northeast to southwest direction) and, thus, the dispersion of pollutants [37]. The land use geography in Mexico City is complex, but as a general description, Rodríguez-Salazar et al. [38] stated that the main industrial center, with a high population density, is located in the northern part of the city. The central part includes the historical and business center of the city, with high commercial activity.

The southern area has been dominated by residential and commercial activities. It was in the 1980s with the liberation of the economy that a process of absolute and relative deindustrialization related to the global economy began in the metropolis [39]. Factories were obligated to settle beyond the limits of Mexico City (formerly called the Federal

District) and the consolidation of light manufacturing began; overall, the tertiarization of the economy led to commercial and services activities beginning to dominate the economy in the city [40].

## 4. Materials and Methods

### 4.1. Data Collection

We collected 482 road dust samples associated with 482 sampling points for this study in a semi-grid pattern of approximately 1-km-wide squares across the urban area of Mexico City (Figure 1). The project was executed only within this political jurisdiction given that this was a project funded by the government. Thus, this sampling can be considered systematic, which covered the urban and peri-urban parts of the city. The collection of samples was done during the dry season in April and May (30 days) of 2017. The atmospheric conditions were stable; the temperature was between 15 and 20 °C, with winds in a north to south direction and speeds between 4 and 8 km/h [41]. During the sample collection campaign, there were no rains. All samples were collected under the same conditions: in a square meter of area on the pavement, below the sidewalk, the distance from the pedestrian area was between 0 and 1 m. All samples were taken in horizontal streets and without sedimentary traps (holes or potholes) to avoid biases. Because the sampling was systematic, the distance to the traffic lights was not considered. The urban road dust was sampled by brushing it from a 1 m$^2$ area at each point. The dust load in each sample point is defined as the amount of dust (<250 μm) per area of street space—in this case, 1 m$^2$—after the coarse material is removed [42,43]. Particles of less than 250 micrometers are most likely to adhere to hands and therefore be involuntarily ingested [44]. This particle size can be easily obtained in the laboratory using a sieve, which is very useful when analyzing a large number of samples.

The concentrations (mg/kg) of 14 elements were determined via inductively coupled plasma optical spectrometer (ICP-OES), which is a methodology used previously to this end [20,45,46]. However, only the most polluting metals identified in a previous study [47] were considered in the present work. Those elements were Chromium (Cr), Copper (Cu), Nickel (Ni), Lead (Pb), and Zinc (Zn). The concentrations were determined by digesting 0.4 g of each sample with 20 mL of concentrated $HNO_3$, using Teflon PFA beakers, in an ETHOS Easy microwave digestion system (Millestone Inc, Milan, Italy). The temperature was brought to 175 $\pm$ 5 °C in approximately 5 min and was kept for 4.5 min. After cooling, the digested samples were filtered with Whatman No. 42 paper, transferred into 50 mL flasks, and graduated with water type A (US-EPA method 3051A). Quality controls for the acid digestion method included reagent blanks and sample duplicates. The quality assurance and quality control (QA/QC) results showed no signs of contamination or loss in any of the analyzes. An Agilent Technologies 5100 ICP-OES (US: EPA method 6010C), sourced from Santa Clara, CA, USA, was used to analyze, in triplicate, the digestions and quality controls. A multi-elemental QCS-26R reference certified material was used (high purity brand) to prepare the calibration curve. The radiofrequency power (RF power) was 1.2 kW, the nebulization flow was 0.7 L/min, and the argon plasma flow was 12.0 L/min.

The pollution load index (PLI) was calculated as the geometric average of the Cr, Cu, Ni, Pb, and Zn results divided by their corresponding background value [48]. The contamination factors were obtained by dividing the concentrations of each heavy metal at each sampling point by the background value. We used the world background values for soils [49], which were obtained by determining heavy metal concentration in soils from places with the least anthropic disturbance possible. A PLI value close to 1 indicates that the heavy metal load was close to the naturally occurring level, while a PLI > 1 indicates contamination [48,50]. In Table 1, we can see the descriptive statistics of the dependent variables. The concentration of Pb in one sample was below the detection limit (Pb = 3.75 mg/kg). We took that limit as the Pb content in the sample because the statistical analysis requires numerical variables.

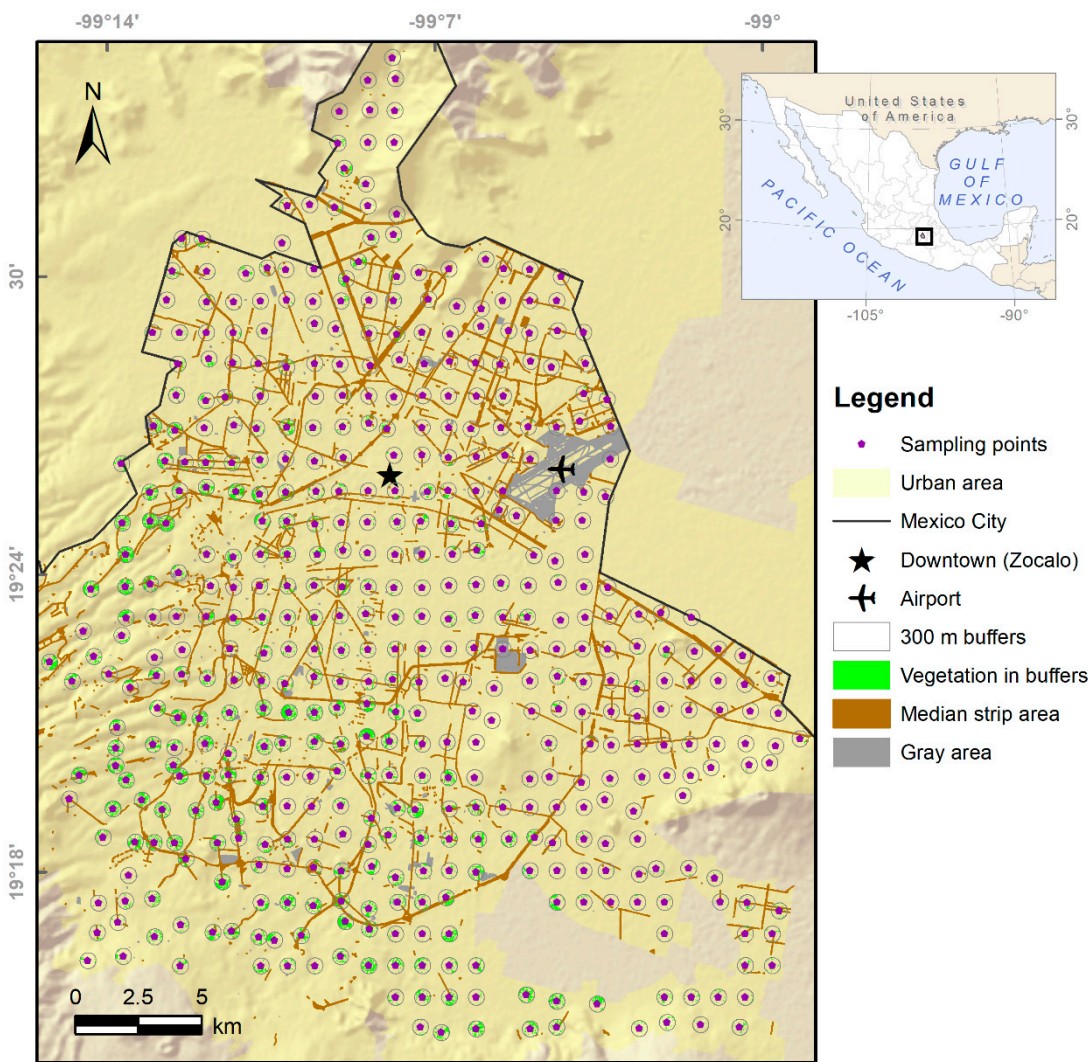

**Figure 1.** The study site and sampling point locations.

**Table 1.** The descriptive statistics of the heavy metal concentrations (mg/kg) in the 482 road dust samples.

| Heavy Metal | Global [1] Background | Minimum | Maximum | Median | Mean | Standard Deviation |
|---|---|---|---|---|---|---|
| Cr | 59.5 | 15.0 | 441.0 | 43.7 | 51.4 | 34.3 |
| Cu | 38.9 | 6.2 | 847.1 | 81.2 | 99.7 | 75.8 |
| Ni | 29 | 13.7 | 148.7 | 35.0 | 36.3 | 13.9 |
| Pb | 27 | 8.8 | 1907.8 | 101.2 | 128.2 | 134.6 |
| Zn | 70 | 18.7 | 4827.6 | 229.9 | 280.7 | 294.4 |
| PLI [2] | | 0.3 | 6.3 | 2.0 | 1.9 | 0.8 |
| Dust load [3] | | 5.4 | 173.3 | 43.0 | 46.4 | 23.2 |

Note: [1] Kabata-Pendias (2011); [2] unitless; [3] Amount of dust <250 μm in 1 m$^2$ of the street (g/m$^2$).

Different explanatory variables were measured at each sampling point (Table 2). The population density was measured as inhabitants per hectare at the census tract level using the 2010 census data from the National Institute of Statistics and Geography [51]. The population density can determine the intensity of various socioeconomic activities that worsen air pollution [8]. Job density considered the number of jobs per hectare within the traffic analysis zone (TAZ) according to the 2017 Household Origin-Destination Survey [52]. Jobs were estimated according to the number of trips to work to a certain TAZ.

**Table 2.** Descriptive statistics of the covariates in the analysis. n = 482.

| Covariate | Minimum | Maximum | Median | Mean | Standard Deviation |
|---|---|---|---|---|---|
| Population density (hab/ha) | 0.0 | 443.1 | 132.4 | 139.8 | 81.1 |
| Job density (jobs/ha) | 4.2 | 349.7 | 30.0 | 43.9 | 50.7 |
| Street intersections | 0.0 | 162.0 | 38.0 | 45.4 | 29.2 |
| Road surface (m$^2$) | 2922.8 | 99,909.4 | 47,475.6 | 47,196.8 | 15,477.1 |
| Distance to the airport (m) | 803.2 | 25,424.3 | 12,792.8 | 12,571.7 | 5329.5 |
| Distance to the city center (m) | 633.9 | 24,361.9 | 10,304.1 | 10,977.3 | 5245.1 |
| Manufacturing units | 0.0 | 377.0 | 13.0 | 15.5 | 21.1 |
| Potentially polluting units | 0.0 | 24.0 | 1.0 | 1.4 | 2.7 |
| Gray area (ha) | 0.0 | 15.7 | 0.0 | 0.3 | 1.4 |
| Entropy index | 0.0 | 0.8 | 0.3 | 0.3 | 0.2 |
| Vegetation (%) | 0.0 | 65.6 | 5.5 | 10.0 | 11.9 |
| Distance to vegetation (m) | 0.0 | 329.3 | 22.0 | 40.2 | 48.2 |
| Median strip area (m$^2$) | 0.0 | 67,855.0 | 1498.0 | 4531.3 | 8161.5 |
| Marginalization index | −1.4 | 1.3 | −0.7 | −0.7 | 0.5 |

Note: A minimum value of zero is expected when the covariate is not present at any of the road dust sampling points.

For the rest of the covariates, we considered a 300 m ring buffer around each sampling site. Street intersections were identified using the street network shapefiles from the official geostatistical framework [51], and, with the network analysis tool in ArcGis 10.0, we counted all intersections within the buffers. The road surface was estimated using this same layer. Street classification is not consistent between the municipalities of Mexico City; thus, for simplicity, we assumed three types of roads and their dimensions: local streets as 7 m wide, intermediate roads as 15 m wide, and highways as 25 m wide. We propose that this classification produces a conservative estimation of the road surface in each buffer.

Based on the 2014 urban geostatistical framework, we calculated the total area of polygons corresponding to large concrete surfaces, such as civil airfields, malls, markets, aviation tracks, and electrical substations. We called these surfaces "gray areas". The median strip area, which is usually covered with vegetation, was estimated using this same dataset, which has polygons explicitly categorized with this name, road median strip. The distance to the city center was calculated using the Euclidean distance (the straight line that connect two points assuming there are no obstacles in the space) from each sampling point to the historical center of downtown (also called Zócalo-Tenochtitlán).

The distance to the airport also considered the Euclidean distance to the Benito Juarez International Airport. The number of manufacturing units in each buffer was calculated based on those classified by the North American industry classification system in 2018. The number of potentially polluting units considered the economic units related to mining, construction, and pipelines in the National Statistical Directory of Economic Units (Directorio Nacional de Unidades Económicas, DENUE) [53].

We calculated the entropy index as a measure of the land-use mix, considering the relative percentages of different land-use types within an area. The entropy index varied from 0 to 1, with 0 indicating a homogeneous area with only one land-use type, and the mix level increasing as the index increases. Here, $P^j$ is the percentage of each land-use type j in the area, and $k \geq 2$ is the number of land-use types j.

$$ENT = -\left[ \sum_{j=1}^{k} P^j ln\left( P^j \right) \right] / ln(k) \tag{1}$$

The entropy index was calculated using information from the publicly available urban data website "Portal de Datos de la Ciudad de México" [54]. Each land-use polygon was georeferenced as a data point, and information about the land use type and the area was provided. Thus, all polygons whose centroid lay within the 300 m buffer were considered

in the estimation. There were 113 categories of land-use descriptions, and two categories were excluded (no zoning—"sin zonificación"—and existing uses—"usos existentes").

Then, the 111 categories were simplified to six general categories: (1) green areas, parks, open spaces, and agricultural areas; (2) residential; (3) office and commercial; (4) industrial; (5) mixed-use; and (6) institutional and public facilities. The official information did not provide concrete and clear definitions of the 111 initial categories; thus, a series of assumptions were made in the collapsing process. For example, land use corresponding to residential and low-scale retail was considered residential. It is very common that, in middle and low-income neighborhoods, small-scale retail coexists with residential uses.

Every original category that includes office and services, was counted in category 3. The mixed-use category included all polygons that were explicitly considered as this on the website as well as those centers of traditional towns and suburbs called "Centros de Barrio". Different land uses, such as residential, commercial, office, and open spaces, converge in these places. The column open area was used for those polygons lacking information about their area. Other corrections were made through a visual inspection on Google Earth for polygons with an important area within the buffer but not initially taken into account when their centroid was not within the buffer.

We calculated the mean normal difference vegetation index (NDVI) in each buffer zone from satellite images (March to May 2017) obtained from Planet Scope for Mexico City to estimate the urban vegetation. A supervised classification was made from the NDVI raster file in the QGIS version 3.4 software, using the "Semi-automatic Classification Plugin" tool. Two classes were determined: (1) vegetation, for the highest NDVI values (>0.6); and (2) no vegetation or the remainder of the urban area. Subsequently, the results of the classification were transformed into a shapefile to obtain the vegetation polygons and to calculate their area.

The vegetation polygons intersecting the buffers were summed up to obtain the total area of vegetation within each buffer. Finally, the total vegetation area was divided by the buffer surface and multiplied by 100 to obtain the percentage of vegetation present in each buffer, referred to here as "vegetation (%)". Another variable related to vegetation is the Euclidean distance from the sampling point to the closest vegetation polygon. Finally, as a measure of the socioeconomic characteristics, the index of marginalization was extracted at the census tract level for each sampling point. This index can have negative and positive values; the highest positive values correspond to the highest levels of marginalization [55].

We considered local variables as those related to characteristics of the immediate urban environment that surrounds the sample point. Examples of these are population density, percentage of vegetation, and the number of potentially polluting units. On the other hand, regional variables were those that characterize a sample with respect to a metropolitan point of reference. Examples are the distance from the city center, and the distance to the airport.

*4.2. Spatial Autocorrelation*

The univariate spatial autocorrelation was examined using Moran's *I* statistics for global measurements of spatial dependence [56]. The formula for standard correlation is expanded to incorporate a spatial weight matrix. Thus, its formula is defined as:

$$I = \frac{n}{S_o} \frac{\sum_{i=1}^{n} \sum_{j=1}^{n} w_{i,j} z_i z_j}{\sum_{i=1}^{n} z_i^2} \tag{2}$$

where *n* is the number of sample points indexed by *i* and *j*, $Z_i$ is the deviation of an attribute (in this case, the heavy metal content) for point *i* from its mean, $W_{i,j}$ is the spatial weight between point *i* and *j*, while So is the aggregate of all spatial weights.

We defined neighbors as those spatial units within a specific distance threshold. For this analysis, we tested different thresholds of contiguity (1600, 5000, 10,000, and 15,000 m). All neighbors were weighted equally and then row standardized. This means that it

depends on the number of neighbors to determine the final weight, as the larger the number, the lower the weights.

The range of Moran's I varies with the weights matrix, but it is usually expected to vary from −1 to 1. The sign corresponds to the type of autocorrelation, i.e., positive or negative. Values close to 1 indicate high spatial autocorrelation while values close to zero mean null spatial autocorrelation. The Moran's test for spatial autocorrelation uses the spatial weights matrix and tests the null hypothesis statement of 'no spatial autocorrelation'. In other words, the alternative hypothesis is if the Moran's I is greater than zero. Then, a *p*-value lower than 0.05 means there is strong evidence to reject the randomization null hypothesis in favor of an alternative hypothesis. We used the function "moran.test" of the R Project program to test the spatial autocorrelation. These results were compared with a Moran Monte Carlo permutations test and also using an inverse distance criterion to determine weights for neighbors. These results were relatively consistent with the moran.test function.

### 4.3. Regression Models

When two variables are close to a perfect linear combination of one another, it is called collinearity. Thus, when more than two variables are involved, it is called multicollinearity. This represents a problem when applying linear regression models given that it can result in unstable regression estimates with high standard errors. Variance inflation factors (VIF) measure how much the variance of the estimated regression coefficient is inflated by the existence of correlation among the predictor variables. The general rule of thumb is that VIFs exceeding 4 warrant further investigation, while VIFs exceeding 10 signify serious multicollinearity requiring correction. Thus, the variance inflation factor (VIF) was used to test the multicollinearity between the factors.

Ordinary least squares (OLS) regressions were tested for all response variables, i.e., the heavy metals contents. Due to a lack of normality evaluated through the Shapiro test, all covariates and dependent variables were log-transformed using a natural logarithm. Some variables with zero values were transformed using the natural logarithm of 1+x. An OLS model can be described in the form:

$$Y = \beta X + \varepsilon \qquad (3)$$

where Y is the dependent variable, X is the independent variable, and β its coefficient. If we consider several covariates, then we have a vector of X and β. Finally, ε is the error term. Coefficients of log-log regression models can be interpreted in the following way, given a coefficient in the form "0.X". We expect about X increase in heavy metal content when the covariate increases by 10%.

After that, a spatial diagnostic of the OLS residuals was applied to select the appropriate spatial regression model for each of these dependent variables [57,58]. There are two main sources of spatial dependence: (1) spatial diffusion, which occurs when spatially proximate units are influenced by the behavior of their neighbors. This was modeled via a Lagrange multiplier spatial lag model (LMlag), estimated by the maximum likelihood; and (2) the geographic clustering of covariates, also called attributional dependence. This was modeled via a spatial error model (LMerror), estimated either with the maximum likelihood or with the generalized method of moments.

The Spatial lag model takes the form:

$$Y = \beta X + \rho W y + \varepsilon \qquad (4)$$

Here, to the OLS equation it is added a W term that stands for the spatial weights matrix and the ρ coefficient.

In the case of the Spatial error model, this takes the form:

$$\varepsilon = \lambda W \varepsilon + v \qquad (5)$$

We can see that this model is related to the error term, where W is also the spatial weights matrix, it is included the lambda coefficient and v is an uncorrelated additional error term.

Lagrange multiplier diagnostics for spatial dependence in R use the regression object and the object of neighbors and weights. In the case of the Spatial lag model, we test for a missing spatially lagged dependent variable, in other words, the null hypothesis is that rho = 0. In the case of the Spatial error model, the null hypothesis is that lambda ($\lambda$) = 0. The first step was to run the non-robust LMlag and LMerror diagnostic tests, the results of which can lead to three different paths: (1) if none of these diagnostic tests are statistically significant, then the OLS estimates are sufficient to model the dependent variable; (2) if only one of the diagnostics determined the presence of spatial dependence, the corresponding model was estimated; (3) if both diagnostic tests were significant, then both the robust LMlag and the robust LMerror diagnostics were tested, and the model with the largest value was used [58]. All the statistical analyses were done with the R Studio program, version 3.6.1.

## 5. Results

The median concentrations of the heavy metals were higher than the global background value in the soils, except for Cr. In the case of Ni, the median concentration is only 6 units above the global background level. Indeed, 91.5% of the samples had a pollution load index (PLI) greater than 1 (median=2) (see above Table 1). We propose that this is an important indication of the anthropogenic origin of the heavy metals found in the road dust from Mexico City. The dust load varied considerably from 5.4 to 173.3 g/m$^2$. As it is very difficult to control aspects related to the deposition of dust in the points of sampling, we decided to work with the heavy metal concentrations instead of the total heavy metal contents as dependent variables.

The mean concentrations of the heavy metals reported here are lower than those reported in a previous study in Mexico City (sampling in 2011) [35]. Even though there are not yet maximum permissible levels in urban road dust, the content of heavy metals in the samples is fortunately still lower than the closest regulation, i.e., the content of heavy metals in soils from residential areas. Therefore, it is important to continue monitoring for long-term fluctuations to avoid an increase in these pollutants and prove the good performance of the environmental policies.

### 5.1. Spatial Autocorrelation

Moran's I was significant for Cr only at a vicinity distance of 5000 m; however, the coefficient of autocorrelation was only 0.03. Therefore, a general pattern of clustering was not expected (Table 3). Cu had a significant Moran's I at all vicinity distances, but, again, the coefficient was very small. Moran's I for Ni became significant at a distance of 10,000 m and was even more significant at 15,000 m; however, the coefficient was only 0.01.

Contrary to Ni, Moran's I for Pb was significant at short distances (1600 and 5000 m), but, again, the coefficients were smaller than 0.1. Zn had a significant Moran's I at all distances, except at 5000 m, with the coefficient smaller than 0.1. Moran's I for the PLI was significant at all distances, and it was greater than 0.1 at 1600 m; hence, a general pattern of clustering could be expected at a short vicinity distance. The dust load had a significant Moran's I at all distances; thus, the coefficients (and consequently the autocorrelation) decreased as the distance increased.

As the global Moran's I was very close to 0 for all heavy metals, indicating low levels of positive spatial autocorrelation, it was not necessary to explore the local version of this index to identify any spatial clustering pattern. This is an initial sign that the local aspects are more relevant than any regional process in determining the concentrations of these metals.

**Table 3.** The global Moran's I and test of statistical significance at different vicinity distances (1600, 5000, 10,000, and 15,000 m).

| Variable | 1600 m | *p*-Value | 5000 m | *p*-Value | 10,000 m | *p*-Value | 15,000 m | *p*-Value |
|---|---|---|---|---|---|---|---|---|
| Cr | 0.05 | 0.07 | 0.03 | 0.00 *** | 0.00 | 0.41 | 0.00 | 0.72 |
| Cu | 0.08 | 0.00 ** | 0.05 | 0.00 *** | 0.02 | 0.00 *** | 0.01 | 0.00 *** |
| Ni | 0.00 | 0.11 | 0.02 | 0.06 | 0.01 | 0.01 * | 0.01 | 0.00 *** |
| Pb | 0.06 | 0.02 | 0.03 | 0.00 ** | 0.00 | 0.14 | 0.00 | 0.49 |
| Zn | 0.06 | 0.01 * | 0.01 | 0.14 | 0.01 | 0.01 * | 0.01 | 0.00 *** |
| PLI | 0.13 | 0.00 *** | 0.06 | 0.00 *** | 0.02 | 0.00 *** | 0.01 | 0.00 *** |
| Road dust load | 0.18 | 0.00 *** | 0.16 | 0.00 *** | 0.10 | 0.00 *** | 0.04 | 0.00 *** |

\* Significance at 95% confidence, ** 99% confidence, and *** 99.9% confidence.

In Table 4, we can see the *p*-values of the diagnostic tests of the OLS residuals. All non-robust tests were non-significant. In the case of the robust versions, for Cu, Ni, Zn, the PLI, and the dust load, the tests were also non-significant, which indicates that OLS is a suitable modeling approach. For Pb, the spatial error model was significant (alpha at 5%); thus, for this metal, the results of the spatial error regression model were analyzed.

**Table 4.** *p*-values of the diagnosis of the ordinary least squares (OLS) regression residuals.

| Test | Cr | Cu | Ni | Pb | Zn | PLI | Dust Load |
|---|---|---|---|---|---|---|---|
| Lm (lag) | 0.11 | 0.15 | 0.51 | 0.51 | 0.89 | 0.65 | 0.50 |
| LM (error) | 0.06 | 0.24 | 0.75 | 0.14 | 0.56 | 0.47 | 0.24 |
| Robust LM (lag) | 0.11 | 0.35 | 0.14 | 0.07 | 0.31 | 0.50 | 0.18 |
| Robust LM (error) | 0.06 | 0.65 | 0.17 | 0.02 * | 0.24 | 0.38 | 0.10 |

\* Significance at 95% confidence.

### 5.2. Factors Influencing the Heavy Metal Concentrations

The VIF indicators were below 4.3 for our vector of covariates, which indicates a low risk of multicollinearity that could bias the coefficient estimations. Overall, there were few significant relationships between the covariates and the heavy metal concentrations, including the road dust load (Table 5). Population density had a null association with all of the heavy metals, which indicates that the number of people in the surroundings of the sample point was not relevant to explain the presence of our dependent variables.

**Table 5.** Results of the ordinary least squares regressions.

| Covariate | Cr | | | Cu | | | Ni | | | Pb | | |
|---|---|---|---|---|---|---|---|---|---|---|---|---|
| | *beta* | *p* | | *beta* | *p* | | *beta* | *p* | | *beta* | *p* | |
| **Intercept** | **4.94** | **0.00** | *** | **3.64** | **0.03** | * | 3.30 | 0.00 | *** | 6.76 | 0.00 | ** |
| Population density (inhabitants/ha) | 0.00 | 0.98 | | −0.03 | 0.40 | | −0.02 | 0.36 | | 0.03 | 0.49 | |
| Job density (jobs/ha) | −0.04 | 0.46 | | 0.11 | 0.10 | . | −0.01 | 0.76 | | 0.04 | 0.61 | |
| Street intersections | 0.02 | 0.71 | | −0.06 | 0.49 | | 0.02 | 0.63 | | 0.01 | 0.93 | |
| Road surface (m$^2$) | −0.07 | 0.48 | | 0.03 | 0.85 | | −0.07 | 0.35 | | −0.09 | 0.59 | |
| Distance to the airport (m) | −0.06 | 0.38 | | 0.05 | 0.57 | | 0.10 | 0.06 | . | −0.18 | 0.11 | |
| Distance to the city center (m) | 0.01 | 0.87 | | −0.04 | 0.71 | | 0.02 | 0.79 | | −0.03 | 0.84 | |
| Manufacturing units | 0.04 | 0.34 | | 0.10 | 0.04 | * | 0.02 | 0.37 | | 0.07 | 0.27 | |
| Potentially polluting units | 0.08 | 0.12 | | −0.03 | 0.62 | | −0.02 | 0.58 | | 0.00 | 0.96 | |
| Gray area (ha) | 0.00 | 0.97 | | 0.08 | 0.45 | | 0.05 | 0.43 | | −0.03 | 0.81 | |
| Entropy index | 0.02 | 0.93 | | -0.17 | 0.46 | | 0.37 | 0.01 | ** | 0.15 | 0.60 | |
| Vegetation (%) | −0.01 | 0.82 | | −0.03 | 0.60 | | −0.05 | 0.10 | | −0.01 | 0.90 | |
| Distance to vegetation (m) | −0.01 | 0.68 | | 0.02 | 0.53 | | −0.03 | 0.21 | | 0.01 | 0.84 | |
| Median strip area (m$^2$) | 0.01 | 0.10 | . | 0.02 | 0.09 | . | 0.01 | 0.09 | . | 0.02 | 0.08 | . |
| Marginalization index | −0.01 | 0.67 | | −0.01 | 0.80 | | 0.01 | 0.80 | | −0.06 | 0.26 | |
| r$^2$ | −0.01 | | | 0.06 | | | 0.02 | | | 0.05 | | |

**Table 5.** *Cont.*

| Covariate | Zn | | PLI | | Dust Load | | |
|---|---|---|---|---|---|---|---|
| | *beta* | *p* | *beta* | *p* | *beta* | *p* | |
| **Intercept** | 2.98 | 0.07 | . | 0.59 | 0.62 | 8.13 | 0.00 | *** |
| Population density (inhabitants/ha) | −0.02 | 0.59 | | -0.01 | 0.77 | −0.01 | 0.81 | |
| Job density (jobs/ha) | 0.08 | 0.22 | | 0.04 | 0.43 | −0.16 | 0.01 | ** |
| Street intersections | −0.11 | 0.16 | | -0.02 | 0.69 | 0.09 | 0.25 | |
| Road surface (m$^2$) | 0.23 | 0.09 | . | 0.00 | 0.98 | −0.30 | 0.02 | * |
| Distance to the airport (m) | 0.03 | 0.76 | | −0.01 | 0.84 | −0.06 | 0.44 | |
| Distance to the city center (m) | −0.03 | 0.78 | | −0.01 | 0.86 | −0.03 | 0.79 | |
| Manufacturing units | 0.08 | 0.10 | | 0.06 | 0.08 | . | −0.04 | 0.43 | |
| Potentially polluting units | −0.03 | 0.65 | | 0.00 | 0.99 | 0.12 | 0.05 | . |
| Gray area (ha) | 0.05 | 0.60 | | 0.03 | 0.69 | −0.01 | 0.91 | |
| Entropy index | −0.15 | 0.51 | | 0.04 | 0.79 | 0.39 | 0.07 | . |
| Vegetation (%) | −0.04 | 0.43 | | −0.03 | 0.46 | −0.13 | 0.00 | ** |
| Distance to vegetation (m) | 0.03 | 0.37 | | 0.01 | 0.84 | 0.01 | 0.77 | |
| Median strip area (m$^2$) | 0.01 | 0.21 | | 0.01 | 0.04 | * | 0.01 | 0.28 | |
| Marginalization index | −0.04 | 0.29 | | −0.02 | 0.41 | 0.02 | 0.51 | |
| r$^2$ | 0.07 | | 0.04 | | 0.10 | | |

beta is the slope; *p* represents the *p*-value; **.** Significance at 90% confidence, * 95% confidence, ** 99% confidence, and *** 99.9% confidence.

Job density had a weak positive association (significance level of 90%) with Cu. However, there was not any significant relationship with the rest of the metals. Our initial expectation was that in places with high job density, the heavy metal concentrations would be high due to an increased number of trips to work and, therefore, increased levels of polluting emissions. Thus, the association found between job density and Cu supports our initial hypothesis since the emission of this metal from vehicles has been associated with tire wear and brake abrasion [29] With the dust load, job density had a significant inverse association (significance level of 99%). We could expect about a 1.6% increase in the dust load when the job density decreased by 10%. This could be explained if we consider that employment centers in Mexico City are related to tertiary types (offices and commerce) that are not necessarily highly polluting activities. These workplaces tend to be better cared for and cleaner than lower-class areas.

The street intersections variable was not significant with any metal. The initial expectation was that this variable could represent the emissions of heavy metals due to car braking, with the higher the number of street intersections representing higher braking frequency. However, the braking emissions are likely not large enough to be detected through street intersections.

The road surface had a weak positive relationship with Zn (significance level of 90%). The emissions of Zn from vehicles have been associated with the combustion of lubricating oil [30], tire wear [25], and diesel exhaust emissions [27]. Therefore, this result supports the expectation that the road surface is a suitable proxy variable for traffic flow as a determinant of Zn in the road dust. On the other hand, there was a significant inverse association between the road surface and the dust load (significance level of 95%). We could expect about a 3% increase in the dust load when the road surface decreased by 10%. These surfaces are likely maintained and cleaned as brigades of cleaning workers sweep the larger roads.

The distance to the airport had a positive but weak relationship with Ni (significance level of 90%). Higher concentrations of this metal were found further away from the airport; therefore, the airport might not be an important source of this metal. An inverse weak association (significance level of 90%) was found for Pb and the distance to the airport in the spatial error model (Table 6). The coefficient tells us that we could expect about

a 2% increase in the Pb content when the distance to the airport decreased by 10%. Our hypothesis is that tire wear in the aircraft take-off and landing could emit dust particles containing Pb. In the case of the distance to the city center, there was a null association with the other heavy metals.

**Table 6.** Spatial error model results for Pb.

| Covariate | Pb | |
|---|---|---|
| | *beta* | *p* |
| Intercept | 7.12 | 0.00 |
| Population density (inhabitants/ha) | 0.03 | 0.41 |
| Job density (jobs/ha) | 0.03 | 0.68 |
| Street intersections | 0.01 | 0.88 |
| Road surface (m$^2$) | −0.10 | 0.56 |
| Distance to the airport (m) | −0.19 | 0.08 |
| Distance to the city center (m) | −0.05 | 0.68 |
| Manufacturing units | 0.08 | 0.19 |
| Potentially polluting units | 0.00 | 0.96 |
| Gray area (ha) | −0.01 | 0.94 |
| Entropy index | 0.17 | 0.54 |
| Vegetation (%) | 0.00 | 0.95 |
| Distance to vegetation (m) | 0.01 | 0.87 |
| Median strip area (m$^2$) | 0.02 | 0.08 |
| Marginalization index | −0.06 | 0.20 |
| AIC | 757.01 | |

Note: AIC (Akaike Information Criterion), lower AIC value suggest a better fit.

Manufacturing units had a positive significant association with Cu with an alpha level of 5%. We could expect about a 1% increase in Cu when manufacturing units increased by 10%. It is very likely that behind this variable there are a variety of chemical processes in the manufacturing; therefore, there could be several sources of Cu from these processes as well as traffic-related emissions. The relationship with PLI was also positive but less significant (significance level of 90%). Potentially polluting units failed to show any association with the heavy metals. This means that possible major pollutant units were not properly identified with our variable. Thus, other alternatives must be tackled in future research such as the use of other classification schemes to differentiate potentially polluting units from the whole census of economic units. Only in the case of the dust load was there a positive significant association at the 90% significance level. Thus, these units are associated with an increase in dust, but are not associated with an increase in the heavy metal content.

The gray area variable also failed to detect any association with the heavy metals. Our initial expectation was that this variable could be related to heavy metals due to the polluting emissions of activities related to large concrete surfaces (markets, aviation tracks, and electrical substations). The null association of the gray areas could be due to the huge diversity of activities considered in the covariate. The entropy index had a positive relationship at the alpha level of 5% with Ni. We could expect about a 4% increase in Ni when the entropy index increased by 10%. A high entropy index means a similar proportion of the six land uses considered; thus, these places could have a variety of potential sources of Ni together, such as sites of fuel combustion.

There was also a weak positive relation (alpha level of 10%) of the entropy index with the dust load. The difficulty in obtaining a clear relationship between the land-use covariates and heavy metal contents in urban dust in México City could be related to the tertiary-oriented economy. We assume that mobile sources of traffic emissions would be more relevant but also more difficult to trace. In the case of air pollution, industrial point emissions have been identified as the source of pollution, such as in the studies of Zhang

et al. [33] in Alberta, Canada, and Jung et al. [8] in Korea, where commercial and industrial areas were associated with increased particulate matter pollution.

The percentage of vegetation in the buffer and the distance to the closest vegetation spot failed to show any association with the heavy metal content. In the case of the former, there was a significant negative association (significance level of 99%) with the dust load, which indicates that vegetated areas tended to have less dust but not necessarily a higher heavy metal content. The median strip area had a weak (significance level of 90%) but consistent positive relationship with Cr, Cu, Ni, Pb, and PLI. This led us to hypothesize that these areas may act as sinks of pollution in the roads, acting as places of heavy metal accumulation. For example, during the sweeping of the road, the dust may be dumped there. The positive relationship between the median strip area and Pb remained for the spatial error model at the alpha level of 10% (Table 6).

Initially, we considered the median strip area part of the vegetation area because it frequently has a vegetation cover. After testing the initial models, the previous vegetation area covaried positively with Pb and Cr. That result was unexpected, and we therefore decided to separate the components of the vegetation covariation and, finally, identified that only the median strip area was positively related to Cr, Pb, Zn, and the PLI. It will be important to untangle the relationship between heavy metals and the median strip area to define the best way to manage such areas. It is important to clarify that covariates of urban form do not necessarily represent specific detailed sources of heavy metals. They represent general characteristics (places) of the urban environment where these pollutants could be being emitted or places of concentration (sinks of pollution (from other mobile or fixed sources of pollution))).

Finally, there was no differential exposition to heavy metals according to the socioeconomic status of households, as we found a null association with the marginalization index. The good news is that road dust is not an important source of exposure to Pb for those in low-income areas, as they are at higher risk of developing health problems [59]. However, attention should be paid to keep the streets clean, because marginalized areas tended to be dustier than middle-income and affluent areas.

The study of the relationships between the urban form and heavy metals is incipient; further investigation is needed to develop a conceptual framework that guides the development of more robust models. The present study is an exploratory analysis in this sense, and we tested the group of covariates that we considered the most relevant. Although some of these variables did not show any association, the inclusion was supported by a deep reflection of what we considered could be the route of heavy metals in the urban environment.

The study of temporal variation in the short term is also important to design better sampling processes that minimize and control the effects of potential confounding factors. For example, natural cleaning mechanisms, such as rain and air currents, and the different practices of street sweeping. From a methodological point of view, we propose to test smaller buffers, which might be at 100 or 150 m around the sampling points, since the characteristics of the immediate surroundings are very important. A better characterization of the urban form might also benefit from more consistent and refined publicly available data regarding the urban environment. The characterization of the local urban environment is key to devising the sampling strategy. Further research lines also include the application of other modeling approaches and inter-city comparisons to test if the phenomena studied here present similar behavior in cities of different sizes and economic conditions. Furthermore, we suggest the inclusion of covariates related to atmospheric processes as well as covariates with more detailed georeferenced data about emission sources.

## 6. Conclusions

According to the global Moran's I, there were low levels of positive spatial autocorrelation in all the heavy metals analyzed. We interpret this as an indication of the greater relevance of the local aspects over regional processes as determinants of the heavy metal

content in urban road dust. Any mapping exercise based on statistical interpolation would not be reliable. A lack of major unique sources of these pollutants could also cause a lack of spatial autocorrelation. In our regression exercise, the most striking finding was that the median strip area in urban roads had a weak but consistent positive relationship with Cr, Cu, Ni, Pb, and the Pollution Load Index. Other significant positive relationships were found for Cu with the manufacturing units, and Ni with the entropy index. More disaggregated indicators would be relevant to unveil the nature of these associations.

Certain variables failed to show any association with the heavy metals, such as the population density, street intersections, distance to the city center, gray area, distance to vegetation, and marginalized areas. Other variables that failed to be associated with heavy metals but showed an association with the dust load were the potentially polluting units (significance level of 90%) and vegetation (significance level of 99%), positive in the former and negative in the latter. The job density (significance level of 99%) and road surface (significance level of 95%) significantly reduced the dust load as well. For Pb, the spatial error model showed the correct specification, unlike the other metals where OLS was found to be appropriate. In this model, distance to the airport had a weak (significance level of 90%) and an inverse association with Pb. This presents an important suggestion to consider this place as a potential source of this metal in urban dust.

Thus, we can conclude that some features of the urban form, as described above, are important drivers of heavy metal pollution in the road dust. A better understanding of how road dust pollution is associated with the urban form will be important to design measures that mitigate the exposure of people to those pollutants.

**Author Contributions:** A.A.: data cleaning, calculations, statistical analysis, wrote the manuscript. D.B.-H.: original idea, data cleaning, calculations, statistical analysis, directed the research, trained the student, and wrote the final version of the manuscript. F.B.: project coordinator, coordinated the chemical analyzes, proposed the idea, and revised the previous texts. A.G.: project coordinator, reviewed the final version of the manuscript. R.C.: collected the urban dust samples and prepared them for analysis. All authors have read and agreed to the published version of the manuscript.

**Funding:** This research was funded by grant number 283135 SEP-CONACYT.

**Institutional Review Board Statement:** Not applicable.

**Informed Consent Statement:** Not applicable.

**Data Availability Statement:** The data about heavy metal content in the sampling points presented in this study are available on request from the corresponding author. This data are not publicly available due to belong to the funding agency. The rest of the data comes from publicly available datasets. This data can be found here: https://www.inegi.org.mx/programas/ccpv/2010/; http://en.www.inegi.org.mx/programas/eod/2017/default.html#Microdata (accessed 28 October 2020).

**Acknowledgments:** We are grateful to Gutiérrez-Ruiz, M.E., Ceniceros-Gómez, A.E., López-Santiago, N.R. for support in the chemical analysis of the dust samples. We would like to thank the four anonymous reviewers for their valuable feedback in the review process.

**Conflicts of Interest:** We declare no conflicts of interest.

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
