# Peer review of "Is the Urban Form a Driver of Heavy Metal Pollution in Road Dust? Evidence from Mexico City"

_atmosphere, doi:10.3390/atmos12020266_

Round 1

Reviewer 1 Report

In this work road dust was collected right from the ground and analyzed for specific trace metals. The trace metal levels, normalized based on average global values found on soil,  underwent a statistical analysis that did not unveal any significant findings other that the area under investigation, even though covers a large fraction of the city of Mexico, it is rather homogeneous with respect to the variables in question.

The fact that no significant findings are reported, in combination with the fact that the data are already used in another publication decreases the significance of this work. However, this work is still worth publishing if some adjustments are applied to the text.

Major Changes

The authors fail to describe their methods adequately on certain occasions.

Please define dust load. Currently the best definition is a footnote on Table 1. I assume the weighed mass of the collected dust normalized by the surface area of collection and sieved to remove coarse material? A reference and adequate description will be appreciated.

The authors used Moran’s I test but fail to describe how they defined the weights of the test. This is critical as the weights define to a large extent the result. Currently I can only assume how weights were defined. They also do not describe the test, which is a major tool in this work. Please add a description of the test briefly explaining to the inexperienced reader how to interpret the results, followed by an analysis on how weights were defined in this work.

Similarly explain the variance inflation factor (VIF) briefly.

LMlag and LMerror are integrated functions of An R package. It is worthwhile mentioning them but without explaining the math behind them the discussion is not futile.

The authors fail to mention which sources are related to which trace metals. As an example broke abbration is generally related to Cu (Sternbeck, Sjödin, & Andréasson, 2002; Schauer et al., 2006) while Zn with the combustion of lubricating oil (Viana et al., 2008). Both are vehicular, but  are related to different parts of the car. This may shed a light on the discussion on these two trace metals.

Minor comments

This text is very well written. However I am wondering about the use of the word antrhropic which is “of or relating to human beings or their span of existence on earth”. I suggest the use of the term anthropogenic which is more related.

From the manuscript which is not numbered: From a social point of view, aspects, such as tax revenue and education level, are associated with a decrease in urban pollution (Liang et al.,2019). Please also mention the Environmental Kuznets Curve (EKC) (Selden and Song, 1994; Cole, 2003). I assume that the discussion in this paragraph is related to which part of the inverse U is country is located at.

Please mention that trace metals concentration may also increase due to natural processes (eg G. S. Bañuelos & H. A. Ajwa, 1999)

General comment

The spatial autocorrelation would probably yield better results if a clustering method was applied such as PCA or k-clustering.

References

Cole, M. A. (2003) ‘Development, Trade, and the Environment: How Robust is the Environmental Kuznets Curve?’ Environment and Development Economics, vol 8, no 4, pp557–579

S. Bañuelos & H. A. Ajwa (1999) Trace elements in soils and plants: An overview, Journal of Environmental Science and Health, Part A, 34:4, 951-974, DOI: 10.1080/10934529909376875

Schauer, J. J., Lough, G. C., Shafer, M. M., Christensen, W. F., Arndt, M. F., DeMinter, J. T., et al. (2006). Characterization ofmetals emitted from motor vehicles. Health Effects Institute.

Seldon, T. M. and Song, D. (1994) ‘Environmental Quality and Development: Is
There a Kuznets Curve for Air Pollution Emissions?’, Journal of Environmental
Economics and Management
, vol 27, pp147–162

Sternbeck, J., Sjödin, A., & Andréasson, K. (2002). Metal emissions from road traffic and the influence of resuspension—results from two tunnel studies. Atmospheric Environment, 36, 4735–4744.

Viana, M., Kuhlbusch, T. A. J., Querol, X., Alastuey, A., Harrison, R. M., Hopke, P. K., Winiwarter, W., Vallius, M., Szidat, S., Prévôt, A. S. H., Hueglin, C., Bloemen, H., Wåhlin, P., Vecchi, R., Miranda, A. I., Kasper-Giebl, A., Maenhaut, W. and Hitzenberger, R.: Source apportionment of particulate matter in Europe: A review of methods and results, J. Aerosol Sci., 39(10), 827–849, doi:10.1016/j.jaerosci.2008.05.007, 2008.

Reviewer 2 Report

The manuscript has an undoubtable interest but a deep revision is needed before considering its suitability to be published in Atmosphere.

There is a lack of description of the methodology. The number of samples is striking. It should be take into consideration the effort for sampling but it would be convenient to add more information about sampling and analysis. When were the samples collected? How long did it take the sampling? Were the meteorological conditions similar during the sampling period? Rain may significantly affect the dust load deposited and its composition, and it has to be clarified if the sampling had been affected by rain.

More information is needed about micro-location of -sampling. Was the sampling always performed at the pavement? Was also performed at pedestrian or other areas? What was the distance to the border of the pedestrian area and the sampling point? Was this distance take into account? Was the distance of the sampling point to the traffic lights and to the street intersections take into account?

Another parameter, not considered in this study, that can be of interest is the number of cars.

As the author concluded, local characteristics may strongly influence the road dust load and composition; therefore, this should be taken into account when devising the sampling strategy.

More information is needed about sample treatment and analysis. How were the samples dissolved prior to the analysis? Was a bulk acid dissolution or an acid leaching? It would be of high interest to analyse not only the selected elements but also other elements that could help interpretation. This not implies a significant additional effort; given that once sampled and dissolved, the analyses by ICP OE toes not require more time. The disposal of a higher number of elements will make possible to apply receptor modelling for source apportionment studies; as performed by Amato el al 2011. This can be a good approach for future works and would provide more information about the origin of road dust.

The introduction section is not well organized. This can be greatly improved. There are relevant papers on road dust that were not included such as those papers by Amato et al; see some references below.

More information about the potential sources of the metals selected could be added in the introduction section: i.e. Pb can be emitted by fuel combustion; Cu can be related to brake wear and industrial processes; Cr, metallurgical activities; Zn, pneumatics; Ni fuel combustion metallurgy. See papers on PMF by Amato et al 2016 and Viana et al 2008, among others

Table 1. In the description you should compare the results with those obtained in other areas

The following sentence is not correct: “The distance to the airport had a positive but weak relationship with Ni (at the 10% level). Higher concentrations of this metal were found further away from the airport; therefore, the airport must not be a source of this metal”; you can conclude that the airport is not an important source of Ni; but you cannot state that it is not a source of this metal.

“Potentially polluting units failed to show any association with the heavy metals” This is probably due to the fact that the potential polluting units were not properly identified.

“Thus, these units are associated with an increase in dust but are not necessarily contaminated.” What do you mean? How you classified “contaminated dust” and not contaminated dust”?

References

Amato, F et al. 2009 Evaluating urban PM10 pollution benefit induced by street cleaning activities. Atmospheric Environment,  http://dx.doi.org/10.1016/j.atmosenv.2009.06.037, 43 (29), 4472-4480

Amato, F et al. 2009. Quantifying road dust resuspension in urban environment by Multilinear Engine: a comparison with PMF2. Atmospheric Environment. 43(17) 2770- 2780.

Amato, F., et al, 2009. Spatial and chemical patterns of PM10 in road dust deposited in urban environment. Atmospheric Environment, 3 (9), 1650- 1659.

Amato F., et al. 2011. Sources and variability of inhalable road dust particles in three European cities. Atmospheric Environment, 45, 6777-6787

Amato, F et al., 2011. Size and time-resolved roadside enrichment of atmospheric particulate pollutants. Atmos. Chem. Phys., 11, 2917-2931

Amato, F., et al. 2016. AIRUSE-LIFE+: a harmonized PM speciation and source apportionment in five southern European cities, Atmos. Chem. Phys., 16(5), 3289–3309, doi:10.5194/acp-16-3289-2016, 2016.

Karanasiou, A., et al. (2014). Road Dust Emission Sources and Assessment of Street Washing Effect. Aerosol and Air Quality Research, 734–743. doi:10.4209/aaqr.2013.03.0074

Amato, F., et al. 2013. Impact of traffic intensity and pavement aggregate size on road dust particles loading. Atmospheric Environment, 7, 2013, 711-717.

Viana, M. et al. 2008. Source apportionment of particulate matter in Europe: A review of methods and results. Journal of Aerosol Science doi:10.1016/j.jaerosci.2008.05.007, 39, 827-849

Reviewer 3 Report

The manuscript of Aguilera et al. 2021 reported the measurements of heavy metals in Mexico city in the dry season of 2017 to determine the impact of urban form on road dust heavy metals. The research topic is interesting but the method and discussions need to improve and clarify to readers. Particularly, the authors cannot treat the influences of anthropogenic emissions and urban form factors to road dust heavy metals as separate parameters that are entirely independent of each other. Lack of in-deep discussions related to the driving factors behind the observation results. The discussions associated with the measurement method (i.e. ICP-OES) need to be strengthened to convince the readers about the suitability of the method used in this study. Following is a list of additional concerns.

Abstract:

Why did study aims need to conduct after estimating the potential anthropogenic sources of the pollutants? How can the authors estimate this?  

I would suggest providing statistical tests and data to support the conclusions.

  1. Introduction:

Paragraph 1: References to support the final statement. In addition, the authors need to explain why increases in urbanization lead to elevated pollution.

Paragraph 2: Provide references to support the statement that “less attention has been paid to road dust”. Why did the authors need to assume the linkage here?

Urban form is one of the possible factors that contribute to urban pollution, all of the possible factors need to discuss here.

Paragraph 3: Please specific and precise here since not all heavy metals are toxic and bioaccumulation. I would suggest reviewing the heavy metals that were investigated in this study.

Why did the authors separate mobile sources to another paragraph? Moreover, the transitions here to emphasize the role of “characteristic of urban environment” are unclear and need to modify.

 I suggest merging the literature review with the introduction to clarify the study aims.

  1. Study area: A figure to show the location of the sampling area should be useful here, especially to show the valley system. In addition, potential anthropogenic emission activity needs to present here.
  2. Materials and Method

“We used 482 sampling points . . .” => Is it means “482 road dust samples associated with 482 sampling points have been collected in this study . . . “

Via ICP-OES => by using ICP-OES method.

How can the authors “replaced” the sample with Pb concentration below MDL?

The authors must provide more details about the method used to determine the heavy metals concentrations in this study including sample preparation/treatment, MDL, QA/QC + SOP of ICP-OES, SRM recovery, etc. 

Why did the author use the values reported by Kabata-Pendias, 2011?

Fig. 1: no need to explain the central and airport location in the figure caption

Any previous study applied the same method as this study to classify the road surface as well as to category the land-use. If yes, authors could cite these studies to support the method.

Give a definition of “Euclidean distance” to clarify the way of estimation.

Support that the distribution of road dust is not homogeneous, the authors should clarify how can they choose a 1m2 representative area for sampling and calculating.

References should be cited for each criterion of the classification method.

Road dust with a size of <250 µm could not ideal to represent the airborne deposited particles. In addition, sample homogenization is an important issue that should be checked with this type of sample.

  1. Results

Paragraph 1: Considering the high S.D. value (Table 1), Ni is not significantly different from that of background value.

Although the authors think the interpolation exercise is meaningless, Geo chemical mapping for heavy metals is still valuable to provide an overview of spatial distribution information and comparison as well. The authors should compare the heavy metal concentrations obtained in this study with other studies worldwide.

5.1 Spatial autocorrelation

Please clarify the local and regional scales that have been mentioned in this section.

Although this study focus on the impacts of urban form to heavy metals in road dust, an investigation of the contributions of emissions sources could be necessary. If the source contribution dominated then it may outweigh the role of urban factors.

Could the authors comment on the possible heavy metals emitted/processed from each covariate in Table 5?

Table 5 is very hard to read because of the overlap of beta and p. The authors need to modify this table to increase the clarity.

Section 5.2

How can the authors estimate the increase in dust load when job density decreased? Similar question for other estimates since R2 of all models is very small (less than 0.1).

Could the authors explain more details why different land-use types within an area can contribute to the variation of heavy metals in street dust? In addition, variation in Ni concentrations is very small (Table 1) and not different from background value, thus discussions associated with this metal is somewhat speculative.

Weak correlations were found for all urban form factors with heavy metals, suggesting the roles of emissions and atmospheric processes?

Again, road dust at size <250 µm is not ideal to link with air pollution.

Could the authors comment on the mechanism associated with the effects of the Median Strip on street dust heavy metals?

Conclusions:

Some of the information presented in this section has been mentioned in the results and discussion. To avoid redundancy, I would suggest authors simplify this section and just leave some key findings here. Discussion/comparison details should be provided in the results and discussion section.

The implication for long-term monitoring here is somewhat unclear.

Reviewer 4 Report

  • Please improve English. The text should be revised by a native English spoken;
  • In page 4 please explain how as the background values for pollutants obtained;
  • In figure 1, the legend is too long. There is also a comment phrase in the legend "We can also see the city center, the airport ..." that is already described by the symbols. Please cut it;
  • In page 5, the reference "portal de datos da ciudad de mexico" is not equally designed compared with the other references all over the text. Please be consistent all over the text references;
  • The tables all over the text should appear before the comments of values related with that table;
  • Comments on values obtained from tables should be made after the presentation of the correspondent table;
  • All over the Results section (5) the text refer phase like "there is a weak positive relation with..." but no value is identified in the phrase. The values that sustain these conclusion should be included in that particular phrase (ex obtained p-value, or other variable);
  • The conclusions section (6) seems more a result discussion section than a conclusion section;
  • The conclusions section (6) is too long and confuse, should be shortened, focusing and emphasising the most important aspects of paper. Conclusions should be short, focusing all the important remarks obtained in the work.
  •  

Round 2

Reviewer 2 Report

Thank you for addressing most of the changes suggested

Reviewer 3 Report

The authors have responded to my comments in detail and I think the manuscript has been significantly improved and could be considered for publication. Just a very minor comment: 

Since the authors acknowledged that all of the relationships in this study are very weak, hence the final conclusion: "Thus, we can conclude that some features of the urban form, as described above, are important drivers of heavy metal pollution in the road dust." is too strong and could be modified like "Thus, we can conclude that some features of the urban form, as described above, might be considered as important drivers of heavy metal pollution in the road dust."